# VERA VERTO: MULTIMODAL HIJACKING ATTACK

## ABSTRACT

The increasing cost of training machine learning (ML) models has led to the inclusion of new parties to the training pipeline, such as users who contribute training data and companies that provide computing resources. This involvement of such new parties in the ML training process has introduced new attack surfaces for an adversary to exploit. A recent attack in this domain is the model hijacking attack, whereby an adversary hijacks a victim model to implement their own – possibly malicious – hijacking tasks. However, the scope of the model hijacking attack is so far limited to computer vision-related tasks. In this paper, we transform the model hijacking attack into a more general *multimodal* setting, where the hijacking and original tasks are performed on data of different modalities. Specifically, we focus on the setting where an adversary implements a natural language processing (NLP) hijacking task into an image classification model. To mount the attack, we propose a novel encoder-decoder based framework, namely the Blender, which relies on advanced image and language models. Experimental results show that our modal hijacking attack achieves strong performances in different settings. For instance, our attack achieves $94\%$, $94\%$, and $95\%$ attack success rate when using the Sogou news dataset to hijack STL10, CIFAR-10, and MNIST classifiers.

## 1 INTRODUCTION

Machine learning (ML) has become a critical component of various applications. Yet, this development has caused the ML models to be increasingly expensive to train. Hence, the training of ML models has transformed gradually to a joint process, e.g., new parties are included in the training of the model either by providing data or computational resources. However, the involvement of these new parties has created new attack surfaces against ML models, e.g., poison and backdoor attacks (Shafahi et al., 2018; Chen et al., 2017). Another recent attack in this domain is the model hijacking attack (Salem et al., 2022a), where the adversary is able to implement their own – hijacking – task into a target victim model. Concretely, the adversary poisons the training dataset of the target model with their own hijacking dataset. The hijacking dataset is first camouflaged for stealthiness to look similar to the target's model dataset. This attack could induce two different risks. The first one is about accountability which is the main threat for hijacking attacks, where the model owner can be framed by the adversary to perform illegal or unethical tasks without knowing. The second one is parasitic computing, where the model owner pays the model maintenance costs, while the adversary uses/offers it for their own application/service for free. On the other hand, the model hijacking technique can also be adapted to compress models, i.e., training a single model for multiple tasks. However, the previous work limits the applicable domains to computer vision (CV) related tasks, even though ML has achieved great success in many domains, e.g., the multiple available translators such as DeepL and Google Translate, and the different face detectors on social media platforms. Moreover, the previous model hijacking attack mandates that the hijacking and original tasks have the same modality. Relaxing this assumption will significantly increase the risks of the model hijacking attack, as the adversary can now target models with different modalities, i.e., more target models exist for the adversary to perform their attack.

In this paper, therefore, we transform the model hijacking attack into a more general *multimodal* setting, i.e., implementing a hijacking task from a completely different domain. More concretely, the adversary can implement a NLP hijacking task into a CV target model, as illustrated in Figure 1. For short, we refer to our attack as the modal hijacking attack. Our modal hijacking attack follows the same threat model as the model hijacking and poison attacks (Jagielski et al., 2018; Shafahi et al.,

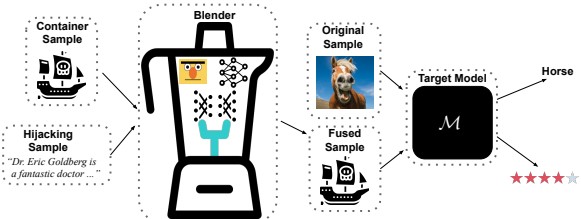

Figure 1: An overview of the multimodal hijacking attack. First, the Blender takes a sample from both the hijacking (a text) and container (an image) datasets. It then mixes both of these inputs to have a fused image with the looks of the container one but with the features of the hijacking text input. The model is able to perform the original classification task (classifying the image as a horse) and the hijacking one, i.e., classifying the fused image as 4-star (the label of the hijacking input).

2018; Sun et al., 2018; Salem et al., 2022a), i.e., the adversary is only able to poison the training dataset without any access to the target model's architecture or hyperparameters. And our modal hijacking attack can induce the same risks as the model hijacking one, i.e., accountability and parasitic computing. Our attack could derive a severer threat to accountability since multiple modalities are involved instead of a single one. And the threat of parasitic computing can be significant when the Blender (introduced in the following soon) is reusable. Different from the previous work, our modal hijacking attack expands the scope from a single type of data to a multimodal setting. The challenge that stems from the change is the needed transformation from a discrete domain (NLP) to a continuous one (CV), in which exists a comprehension gap. We believe this transformation is not trivial, as demonstrated in Section 4.4. Besides, our modal hijacking attack is more general, i.e., the Blender can be applied with different hijacking and original datasets. Hence, it is cheaper for the adversary to hijack target models as shown later in Section 4.4. To the best of our knowledge, the modal hijacking attack is the first work to combine different data modalities, which increases the capability and flexibility for hijacking attacks. Moreover, this work can encourage the exploration of the applicability of performing hijacking attacks for different modalities (which can also be with the benign aim of model compression).

To perform the modal hijacking attack, the adversary needs to transform the NLP-based *hijacking dataset* into the victim model's CV-based *original dataset*. To this end, we propose the Blender, an encoder-decoder-based model which integrates a language model, i.e., BERT (Devlin et al., 2019) and multiple CNN models together. The Blender integrates two losses, i.e., visual and semantic losses, such that it fuses both the hijacking and original inputs to create an output that has a similar visual appearance to the original inputs, while maintaining the features of the hijacking one (as shown in Figure 1). A successful modal hijacking attack should enable the target victim model to preserve its utility, i.e., has the same performance on the original CV task, while performing the hijacking NLP task with high accuracy.

To evaluate our modal hijacking attack, we use two NLP datasets (Zhang et al., 2015), namely Yelp Review (Yelp) and Sogou News (Sogou), and three CV datasets, i.e., MNIST (MNI), CIFAR-10 (CIF), and STL-10 (Coates et al., 2011). We extensively evaluate the different setups for our modal hijacking attack. Our results show that our modal hijacking attack can achieve strong performances with respect to both the attack success rate and victim's model utility. For instance, when victim models trained on MNIST, CIFAR-10, and STL-10 datasets are hijacked by the Yelp (Sogou) datasets, our modal hijacking attack achieves attack success rate of 65% (94%), 68% (94%), and 65% (95%), respectively. Meanwhile, the victim models' utility is not jeopardized, i.e., our modal hijacking achieves the utility of 99% (99%), 93% (93%), and 93% (92%), respectively, which is less than 2% drop compared to clean models. Moreover, we show the generalizability of our modal hijacking attack by evaluating it against different setups, e.g., different models to construct the Blender and target model. Finally, we explore two possible defenses against the modal hijacking attack.

In addition, due to the space limitation, we discuss the limitations of our work in Appendix A for a better understanding.

## 2 BACKGROUND AND RELATED WORKS

**Training Time Attacks:** The inclusion of new parties in the training pipeline for ML has induced a new attack vector. Adversaries can utilize this to interfere with the training of the victim model. These attacks are referred to as training time attacks. One of the most famous training time attacks is data poisoning (Sun et al., 2018; Shafahi et al., 2018; Bojchevski & Günnemann, 2019; Tolpegin et al., 2020; Zhang et al., 2020; Schuster et al., 2020; Carlini & Terzis, 2021). This class of attacks allows the adversary to insert malicious samples into the victim model's training data. One widespread target for the poisoning attacks is to jeopardize the model's utility, i.e., to make the training of the victim model fail, which is different from our modal hijacking attack.

Another related training time attack is the backdoor attack (Yao et al., 2019; Saha et al., 2020; Zhao et al., 2020; Chen et al., 2021; Salem et al., 2020). In this attack, the adversary takes a further step and tries to associate a malicious behavior – of the target model – with a trigger, e.g., a white square on the corner of the input. A successful backdoor attack results in a victim model which behaves benignly on clean inputs; while predicting a specific label when queried by backdoored data, i.e., inputs with triggers, but triggers are not as flexible as our modal hijacking attack.

**Testing Time Attack:** A similar attack is adversarial reprogramming (Elsayed et al., 2019). In this attack, the adversary also tries to perform their own task using a victim model. However, this is a testing time attack, i.e., the adversary only accesses the model after its training. Unlike our hijacking attack, this attack requires assumptions on the target model such as white-box access.

## 3 MODAL HIJACKING ATTACK

### 3.1 THREAT MODEL

In this paper, we generalize hijacking attacks to a multi-modal setting, which represents a more practical scenario in the real world. It might be hard for the adversary to find a victim model of the exact same task domain. Thus, the previous model hijacking attack, which only focuses on CV-related tasks, limits the applications of hijacking attacks. However, the transformation from a model hijacking attack to a multi-modal setting is a challenge. Specifically, NLP tasks locate in a discrete domain while CV tasks are in a continuous one. In that case, how to understand the discrete information from a continuous form is not trivial. To address this challenge, we propose the modal hijacking attack which enables the adversary to hijack the victim model of CV tasks by NLP tasks, indicating a larger scope of applications of hijacking attacks.

We follow the same threat model as the model hijacking attack (Salem et al., 2022a) for our modal hijacking attack, i.e., we only assume the ability to poison the training dataset of the target model. In other words, our modal hijacking attack does not require any extra information about the target model architecture or hyperparameters. This setup is also widely used for poisoning (Jagielski et al., 2018; Shafahi et al., 2018; Sun et al., 2018; Tolpegin et al., 2020) and backdoor (Yao et al., 2019; Saha et al., 2020; Chen et al., 2021; Salem et al., 2022b) attacks.

Moreover, we assume the adversary to have a container – image – dataset to fuse with the hijacking one. This container dataset does not have to be from the same distribution as the original training dataset of the victim model. However, the adversary can construct it depending on their preference for the visual appearance of the Blender's output, i.e., the fused dataset.

Finally, as victim models are used to perform the hijacking task, our modal hijacking attack assumes that the number of labels of the original dataset is at least equal to the hijacking dataset's one.

### 3.2 DATASETS TERMINOLOGY

The modal hijacking attack uses four different datasets which we define now for clarity: First, the *Original Dataset ($\mathcal{D}_o$)*. This is the victim model's training dataset for training the original task. Second, the *Hijacking Dataset ($\mathcal{D}_h$)*. $\mathcal{D}_h$ is the adversary's training dataset for training the hijacking task. Third, the *Container Dataset ($\mathcal{D}_c$)*. $\mathcal{D}_c$ is a set of images the adversary constructs/collects to fuse with the hijacking dataset samples. Finally, the *Fused Dataset ($\mathcal{D}_f$)* which is the container dataset after being fused with the hijacking one.

### 3.3 BLENDER

Intuitively, the Blender aims at generating a fused dataset, which is used for hijacking the victim model. This is performed by fusing the – text – hijacking dataset with the container one. We first present the design of our Blender, then how it is operated and trained.

**Design:** To fuse the text hijacking dataset with an image container one, we first extract the hijacking dataset's features. To extract these features, we follow state-of-the-art works by using a language model(Devlin et al., 2019; Sun et al., 2019). Then to construct the fused dataset, we first try the naive approach of building an adapter, which is a CNN, to resize the NLP features to the size of the victim model's input. However, this approach does not perform well for some datasets as will be shown later in Section 4.4. Moreover, using this naive approach results in random-looking images as illustrated in Figure 6 (Appendix B), which can be easily detected.

To circumvent the limitations of the naive approach, we follow (Salem et al., 2022a) to use an encoder-decoder-like model. More concretely, the Blender consists of an NLP feature extractor $\mathcal{F}_{NLP}$, i.e., a language model, an adapter $\mathcal{A}$, two encoders $\mathcal{E}_1$ and $\mathcal{E}_2$, and a decoder $\mathcal{E}^{-1}$.

Another design decision we make is to use the complete embeddings of the hijacking sentence instead of the last – "[cls]" – token. As will be shown later in Section 4.4, using all of the embeddings significantly improves the performance of the modal hijacking attack.

Finally, our last design choice is to fine-tune the NLP feature extractor on the hijacking dataset before using it. A fine-tuned model is able to understand the specific – hijacking – dataset better. Thus the Blender can better learn the semantic information and fuse it into container images; we later evaluate the performance gain of this step in Section 4.4. It is important to note that this step does not require any additional assumptions since the adversary is the owner of the hijacking dataset, and the NLP feature extractor is completely independent of the target victim model.

**Operation:** We now explain how Blender operates. Firstly, the Blender uses the NLP feature extractor ($\mathcal{F}_{NLP}$) to extract the features/embeddings of the text hijacking input ($x_h \in \mathcal{D}_h$). Next, these features are input to the adapter ($\mathcal{A}$) to preprocess them before being input to the first encoder ($\mathcal{E}_1$). In parallel, the container image ($x_c \in \mathcal{D}_c$) is input to the second encoder ($\mathcal{E}_2$). Next, the outputs of both encoders are concatenated together and input to the decoder ($\mathcal{E}^{-1}$). Finally, the decoder constructs the output fused image ($x_f$), which has the visual appearance of the container image ($x_c$), while having the features of the text one ($x_h$). More formally,

$$\mathcal{E}^{-1}\left(\mathcal{E}_1\Big(\mathcal{A}\big(\mathcal{F}_{NLP}(x_h)\big)\Big)\Big|\Big|\mathcal{E}_2(x_c)\right) = x_f,$$

where $\Big|\Big|$ is the concatenation operator.

**Training:** To train the Blender, we use two losses, namely the visual and semantic losses.

*Visual Loss:* Intuitively, the visual loss ($\mathcal{L}_v$) is the one responsible for forcing the fused image to have a similar look compared to the container one. To accomplish this, we utilize the mean squared error MSE (Jagielski et al., 2018; Cong et al., 2022) to construct the visual loss. More concretely, we compute the pixel-wise difference between the fused and container inputs, i.e., $\mathcal{L}_v = ||x_f - x_c||_2^2$.

*Semantic Loss:* The semantic loss $\mathcal{L}_s$ is designed to fuse the NLP features in the container image. Similar to the visual loss, we use MSE for the semantic loss too. However, the MSE here is calculated between the features extracted from the text input with the ones extracted from the fused image. Feature extraction here is performed with different feature extractors according to the input type, i.e., the text/image feature extractor ($\mathcal{F}_{NLP}/\mathcal{F}_{CV}$) is used to extract the $x_h/x_f$ features. Since the MSE expects the same sizes for both inputs, we further process the adapter's output with a linear layer ($\mathcal{F}_l$) for adjusting its size to match the CV ones. More formally, $\mathcal{L}_s = ||\mathcal{F}_l(\mathcal{A}(\mathcal{F}_{NLP}(x_h))) - \mathcal{F}_{CV}(x_f)||_2^2$.

We use both losses and train the Blender as follows:

1. The adversary first constructs their container dataset $\mathcal{D}_c$. The only requirement for this dataset is to be an image dataset. Ideally, this dataset should have a similar visual appear-

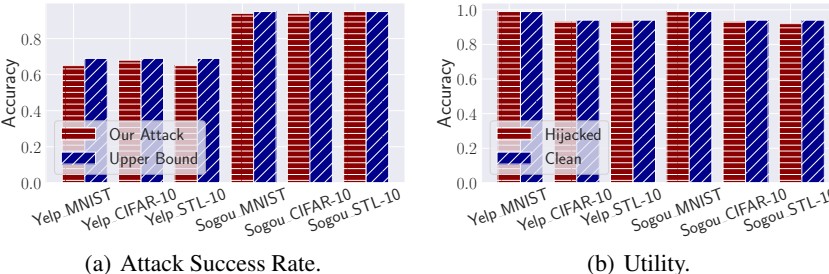

(a) Attack Success Rate.        (b) Utility.

Figure 2: Our multimodal hijacking attack performance. We use x_y notation in the axis to denote the hijacking dataset x and the original dataset y.

     ance as the original dataset $\mathcal{D}_o$, to make the modal hijacking attack more stealthy. However, the adversary can construct this dataset as they desire.

2. Second, every sentence in the hijacking dataset is randomly mapped to a container image from the container dataset. This mapping does not have to be one-to-one, a single container image can be mapped to multiple sentences.

3. Next, each (sentence, image) pair is processed by our Blender as previously presented; and both losses, i.e., semantic and visual, are calculated.

4. Finally, both losses are added and the Blender is updated accordingly, i.e., $\theta = \mathrm{argmin}_\theta(\mathcal{L}_v + \mathcal{L}_s)$, where $\theta$ is the parameters of the Blender and the linear layer $\mathcal{F}_l$.

It is important to mention that the CV feature extractor $\mathcal{F}_{CV}$ and the linear layer $\mathcal{F}_l$ are only needed when training the Blender, then they can be discarded.

### 3.4 THE MODAL HIJACKING ATTACK

The modal hijacking attack is executed in two phases.

**Phase 1:** The adversary starts by training the Blender as previously presented in Section 3.3. Next, they use the Blender to create the fused dataset, i.e., by fusing the container and hijacking datasets. They then perform a label mapping between the original and hijacking dataset. For instance, randomly mapping each label in the hijacking dataset to a distinct one of the original dataset. The adversary then uses this label mapping to decide the labels of the fused dataset, i.e., by mapping the corresponding hijacking samples' labels to the original dataset's ones. Finally, the fused dataset with its labels is used to poison the victim model.

**Phase 2:** After the victim model is trained, the adversary executes the modal hijacking attack on a target hijacking input $x_h$ as presented in Figure 1. As the figure shows, the adversary first samples a container image and then uses the Blender to fuse it with the hijacking input and create the fused image. The adversary then queries the fused image to the victim model and receives the output label. Finally, they map the received label back to its corresponding one in the hijacking dataset.

## 4 EVALUATION

### 4.1 DATASETS DESCRIPTION

As our attack uses datasets from different domains, we start by presenting the computer vision and then the natural language processing related datasets.

**CV Datasets:** We use three commonly-used benchmark datasets (all with 10 classes) in our evaluation, i.e., MNIST, CIFAR-10, and STL-10. *MNIST* is a handwriting digits datasets, which contains 70,000 $28 \times 28$ gray-scale images; *CIFAR-10* is a real-world objects dataset, which contains 60,000 $32 \times 32$ color images; Finally, *STL-10* is also a real-world objects dataset, which has some common (e.g., airplane, cat, and dog) classes with CIFAR-10. We use the labeled subset of STL-10 that

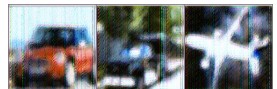 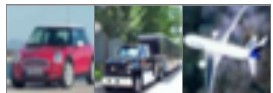

(a) The fused images.      (b) The container images.

Figure 3: The visual results of the Blender's output using the hijacking sentence "I love this place! The food is always so fresh and delicious. The staff is always friendly, as well." Here, the labels for container images are "Automobile", "Truck" and "Airplane", meanwhile, the labels of fused images is "Cat" corresponding to the ground truth of the NLP sentence, i.e., "5 stars".

consists of 13,000 $96 \times 96$ color images. We follow (Salem et al., 2022a) and rescale the images to $3 \times 224 \times 224$ as we are using public CV models as our feature extractors and target models.

**NLP Datasets:** For our NLP datasets, we use two well-established ones, i.e., Yelp Review (Yelp) and Sogou News (Sogou). *Yelp* is a dataset of English reviews with labels corresponding to scores (between 1 and 5). It includes 650,000 training and 50,000 testing samples. *Sogou* is a dataset of news articles with labels associated with five categories, i.e., sports, finance, entertainment, automobile, and technology. It includes 90,000 training and 12,000 testing samples.

## 4.2 EVALUATION SETTINGS

### 4.2.1 MODELS ARCHITECTURE

**Blender:** To recap, our Blender is composed of an NLP feature extractor, an adapter, two encoders, and a decoder, which is depicted in Figure 20 (Appendix E). We now present the architecture of each of these components:

*NLP feature extractor.* We use the Bidirectional Encoder Representations from Transformers (BERT) language model (Devlin et al., 2019), and fine-tune it. Then we discard the last layer, and use the embeddings of each token as our NLP features (as previously described in Section 3.3). We also try different language models and present the results in Section 4.4.

*Adapter.* The adapter is composed of an average pooling layer, and 4 convolutional ones.

*Encoders.* We use the same architecture for both encoders. Each encoder consists of four convolutional layers with batch normalization and ReLU activation function.

*Decoder.* The decoder consists of four convolutional transpose layers. The first three use batch normalization and ReLU activation function, while the fourth one only uses a $\tanh$ activation function.

**CV Feature Extractor:** We adopt the VGG11 (Simonyan & Zisserman, 2015) model as our CV feature extractor and use its output as the features. We evaluate the attack performances when using other CV feature extractors in Section 4.4.

**Victim/Target Model:** We use the ResNet18 (He et al., 2016) model for our evaluation, however, we show the generalizability of our modal hijacking attack with different target models in Section 4.4.

### 4.2.2 EVALUATION METRICS

We follow the same evaluation metrics introduced in (Salem et al., 2022a) which we present below:

**Utility:** We use utility to measure the performance of the victim model's original task. To this end, we train a target model on clean data only. Then we use a clean testing dataset to evaluate its performance on the clean and victim models, where the clean testing dataset is the testing dataset of the original CV model, i.e., from the same distribution as the original dataset but not used when training. We use this dataset to measure the utility when the hijacked model is not jeopardized. And the closer the performances of these two models – on the clean test dataset –, the better the utility is.

**Attack Success Rate:** We use attack success rate (ASR) to measure the effectiveness of the modal hijacking attack, i.e., the performance of the hijacking task. We first create a hijacking test dataset by fusing a clean testing dataset (for the hijacking task) with the container dataset using the Blender.

Next, we compute the accuracy of the victim model on that hijacking test dataset. Moreover, we train an NLP classification model for each hijacking task to compare its performance with the victim model on the hijacking task.

Regarding the aforementioned NLP model, it is trained with the hijacking dataset as is, i.e., the data and labels of the hijacking dataset are not changed. This model represents the case where the adversary wants to perform the hijacking task without the modal hijacking attack, i.e., the upper bound of the attack. The closer the performance of our modal hijacking attack to this model is, the better the attack is. We compare the performance of the NLP and the hijacked models using a hijacking testing dataset from the same distribution as the hijacking dataset. Of course, for our attack, we need to first fuse the hijacking testing dataset with container images using our Blender.

## 4.3 RESULTS

To recap, the adversary first needs to construct a container dataset to train the blender, as previously mentioned in Section 3. We randomly sample 100 images from each target dataset to construct their corresponding container datasets. These 100 images are then removed from the target datasets, i.e., $\mathcal{D}_c \cap \mathcal{D}_o = \Phi$. Next, we sample 5,000 samples from each hijacking dataset, with 1,000 instances per label. For each hijacking-original dataset pair, we randomly map the corresponding 5,000 hijacking samples with the 100 container images and train the Blender as mentioned in Section 3.3.

After training the Blender, we use it to fuse the 5,000 hijacking samples and poison the victim model. The fused samples are combined with the complete training dataset for each victim model, namely 60,000, 50,000, and 5,000 samples for the MNIST, CIFAR-10, and STL-10 datasets, respectively.

To evaluate the performance of our modal hijacking attack in terms of both utility and attack success rate, we train clean models for all original and hijacking tasks. For the original tasks, we use the complete original dataset when training the models. And we use the whole clean testing dataset to evaluate the performances. As mentioned in Section 4.2.2, the closer the performances – with respect to the clean testing dataset – of the hijacked and clean models are, the better the modal hijacking attack is. For the hijacking tasks, we calculate the upper bound of the performance by using the complete training datasets, not just the 5,000 sentences used to hijack the victim model. We then sample a testing dataset from the hijacking's task test dataset and use it to evaluate these models. Finally, we fuse this dataset and use the fused version of the dataset to evaluate the attack success rate of the victim model. The closer the attack success rate to the upper bound performance, the better the modal hijacking attack is.

We first quantitatively evaluate our modal hijacking attack. We plot the attack success rate (ASR) in Figure 2(a). As the figure shows, our modal hijacking attack achieves strong performance independent of the hijacking and the original datasets. For instance, our attack achieves $94\%$, $94\%$, and $95\%$, when using the Sogou dataset to hijack MNIST, CIFAR-10, and STL-10 victim models, respectively, which is only $1\%$ worse than then upper bound models. Similarly, for the Yelp dataset, our attack's ASR is only $4\%$, $1\%$, and $4\%$ less than the upper bound for the MNIST, CIFAR-10, and STL-10 victim models, respectively. This clearly demonstrates the effectiveness of our modal hijacking attack. Specially taking in consideration that we only use 5,000 hijacking sample to hijack the victim models unlike the full dataset when training the upper bound ones. Next, we evaluate our modal hijacking attack's utility. We plot the performances of the victim models and the ones trained with clean datasets in Figure 2(b). As the figure shows, our modal hijack attack achieves almost the same performance on the clean testing dataset compared with the clean models. For instance, the victim model achieves the utility of $99\%$ ($99\%$), $93\%$ ($93\%$), and $93\%$ ($92\%$) on the MNIST, CIFAR-10, and STL-10 models when being hijacked by the Yelp (Sogou) dataset, respectively. This shows the negligible drop in model utility for our modal hijacking attack. In addition, we evaluate the attack performance on the Tiny ImageNet (Deng et al., 2009) as the original dataset. The results show that our attack is generalizable to higher-quality images, as shown in Figure 19 (Appendix D). Besides, we depict some examples with details of the Blender's outputs of Tiny ImageNet in Table 2 (Appendix D)

Second, we quantitatively evaluate the performance of our attack. To this end, Figure 3 shows randomly sampled fused samples together with their container images; when using the Sogou and STL-10 as the hijacking and original datasets, respectively. As the figure shows, the output of our Blender is very similar to the original dataset, with some visible artifacts. However, as shown later (Section 4.5) these artifacts do not make the fused images easier to be detected. Moreover, increasing

the training epochs of the Blender to 200 reduces the artifacts as shown in Figure 9(b). Moreover, to compare the performances of the modal hijacking attack with the naive approach, i.e., only using the adapter not the Blender, we plot the outputs of the adapter of the Yelp samples in Figure 6(a) - Figure 6(f) (Appendix B). Comparing both figures, the output of our Blender is clearly more similar to the original dataset, hence, showing the stealthiness of our modal hijacking attack. We present more examples of the fused dataset and corresponding explanation in Figure 18 and Table 1 (Appendix D).

## 4.4 HYPERPARAMETERS/DESIGN DECISIONS

Due to space limitations, we summarize our findings for exploring different hyperparameters and design decisions of our attack here and present the full evaluation results and additional discussion in Appendix B.

**Using Different NLP Features:** In this section, we evaluate the necessity of using the embeddings of all tokens instead of the single "[CLS]" token. According to our evaluation results, as plotted in Figure 4, our attack showcases the necessity of using all embeddings with an advantageous performance gap.

**Naive Attack:** To show the necessity of our Blender, we evaluate a naive approach for the modal hijacking attack, i.e., directly using the output of the adapter instead of the Blender. Both the quantitative (in terms of ASR and utility as shown in Figure 5) and the qualitative (visual results as shown in Figure 6(a) - Figure 6(f)) results show the effectiveness of our Blender.

**Different Feature Extractors:** We also study the generalizability of our modal hijacking attack by using different feature extractors. From the results shown in Figure 7, we can see that changing the feature extractors yields similar performance, which indicates that our modal hijacking attack is independent of feature extractors.

**Different Victim Models:** We now study the generalizability of our modal hijacking attack against different victim models. From the results shown in Figure 8, we can see that our modal hijacking attack is independent of the victim's model architecture.

**Number of Training Epochs for Blender:** We evaluate the effect of varying the number of the Blender training epochs using the Yelp-CIFAR-10 setting. We train Blenders using from 50 to 500 epochs with steps of 50, then evaluate our attack performance on them. Our results show that the utility and ASR do not change much with the number of epochs (Figure 9(a)). However, the quality of the fused images does get better with a higher number of epochs. The visual quality saturates at approximately 200 epochs. Hence, we believe the adversary can already use 50 epochs if they care less about the visual appearance of the fused images, else 200 epochs would be a good compromise. We show a set of randomly sampled fused images for the different epochs in Figure 9(b).

**Effect of Fine-tuning BERT:** We evaluate the effect of fine-tuning the NLP feature extractor ($\mathcal{F}_{NLP}$). As Figure 10 shows, it is necessary to fine-tune $\mathcal{F}_{NLP}$ with the hijacking dataset. Even the non-fine-tuned one with more poisons, is not comparable to a fine-tuned NLP feature extractor.

**The Poisoning Rate Effect:** To better understand the adversary's capacity, we evaluate the influence of the poisoning rate on our attack. As shown in Figure 11, increasing the poison number indeed improves the ASR, but there exists a saturation. Thus, for the trade-off between computational cost and attack performance, we use 5,000 hijacked samples for our evaluations in Section 4.

**Reusability of the Blender:** As the Blender can be expensive to train, we evaluate reusing a trained Blender to hijack different settings. We try four different setups: the first hijacks models using the same container dataset but different victim models, the second increases the flexibility and use a different container dataset, and the third and fourth further increase the efficiency of the attack by using a pre-trained Blender to camouflage different hijacking datasets. For the first case, we already presented its results when evaluating our attack against different target/victim models, i.e., AlexNet. For the second case, we use the Blender trained using the Yelp – hijacking – and CIFAR-10 – container – datasets to hijack MNIST and STL-10 models. In other words, we use the CIFAR-10 trained Blender to fuse images from the MNIST and STL-10 datasets. Our results show that the modal hijacking attack still achieves strong performance. More concretely, it achieves the same utility compared to using their original Blender (Section 4.3) with an ASR drop of only 2% and 3%

for the MNIST and STL-10 hijacked models, respectively. We present the full results in Figure 12. For the third case, we use the Blender trained using Yelp (Sogou) but conduct the attack with the hijacking dataset of Sogou (Yelp). In that setting, our modal hijacking can gain almost the same performance. For instance, when the adversary trains a Blender using the Yelp (Sogou) dataset to hijack a CIFAR-10 model using the Sogou (Yelp) hijacking datasets, our attack achieves an ASR of 94% (66%) and utility of 93% (94%), which is only 1% lower compared to training a specific Blender for each hijacking dataset. We present the full results in Figure 13. For the final case, we try a different setup where two hijacking datasets (Yelp and Sogou) are used to build a Blender. Then this Blender is used to attack a third dataset (CoLA (Wang et al., 2018), which is a binary dataset). This approach shows that it is indeed better to use two datasets instead of one when building the Blender. More concretely, using the Blender trained jointly on both datasets (Yelp and Sogou) improves the performance to 80% ASR and 93% utility, which is 4% (3%) stronger in ASR than when using a Blender trained on a single dataset, e.g., Yelp (Sogou). We present the full results in Figure 14. These results demonstrate that a single Blender is reusable to different setups, which significantly reduces the cost and increases the flexibility of our modal hijacking attack.

**Container Dataset Creation:** Finally, we propose a way of constructing the container dataset. So far, we use a randomly selected container dataset. One problem is that the labels might not align with the fused samples. Hence, a manual inspection can raise some flags. To this end, we now propose a more stealthy way of constructing the container dataset. We first train a CV classifier, e.g., VGG11, on the container dataset. Next, we sort the images that are misclassified with the most confidence. Finally, we – manually – select the container dataset out of these images. Figure 15 shows a subset of these images. As the figure shows, it is hard to manually assign a label to these images. In other words, it makes the fused samples more stealthy.

## 4.5 DEFENSE

We evaluate using two established defenses against data poisoning attacks to mitigate our modal hijacking attack (Steinhardt et al., 2017; Yang et al., 2022). The first defense, intuitively, clusters a clean dataset and computes the centroid of each label. Then for any given input, the distance between it and its corresponding centroid – depending on its label – is calculated. Inputs with large distances are then discarded. The results consistently show that the performance of the attack drops to almost random guess, however, it induces an average drop of the utility of around 15%. Moreover, this type of defense requires access to clean data which is sometimes hard to get for many applications. The second defense aims to find effective poisons and drop them in low-density gradient regions during training. The results show that the defense is able to reduce the attack success rate, however, our hijacking attack is still effective. For space restrictions, we present the full details in Figure 16 and Figure 17 (Appendix C).

## 5 CONCLUSION

Model hijacking attack is a new threat that makes use of the inclusion of new parties in the ML training pipeline. In this attack, the adversary can hijack CV-based models to implement their own image classification task. However, as the ML has improved into multiple domains besides the CV, the hijacking and original tasks might be from different data modalities. In the paper, we propose a more general *multimodal* hijacking attack, where the adversary can hijack CV models using text classification tasks. To this end, we propose and use an autoencoder-based model that mixes language models with CNNs, namely the Blender, to perform the modal hijacking attack. Using the Blender, the adversary can hijack image classification models using text/NLP hijacking tasks. We extensively evaluate our attack using five different datasets, including three image classification datasets and two text ones. Our results show that the modal hijacking attack achieves strong performances with a negligible drop in the model's utility.

We aim by this work to first raise awareness of the possible accountability risks in some of the realistic machine learning training pipelines. Second, to motivate the community to work on different mitigation techniques to address this risk, we already present a couple of possibilities in that direction. Finally, our modal hijacking technique can also be used for compressing target models, hence reducing their training or maintenance costs.

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

## A    LIMITATION

**Artifacts in the Fused Images:** One limitation of our modal hijacking attack is the visible artifacts in the fused images. We plan to reduce these artifacts by adding a GAN-like discriminator (Goodfellow et al., 2014) to the training of the Blender in future work. This discriminator is trained to differentiate between the container and fused samples. The Blender is penalized for any container sample that the discriminator can identify, hence improving the appearance of the fused samples.

**Computational Costs of Training the Blender:** Another concern is the computational cost of training the Blender. However, as previously mentioned in Section 4.4, our Blender is trained once and can then be used for multiple hijacking attacks even with different hijacking tasks.

**Regarding Mutilple Modalities:** Intuitively, our technique can be adapted for different scenarios where the hijacking task has feature extractors, e.g., language models for the NLP tasks. However, different modalities have different difficulties, i.e., how easy/hard the original data can be modified. For instance, consider hijacking an NLP model with a CV task, which is left for future work. It is indeed an interesting next step but would be significantly difficult/different, as text inputs are not continuous and hence cannot be changed as the images do.

## B    HYPERPARAMETERS/DESIGN DECISIONS

**Using Different NLP Features:**

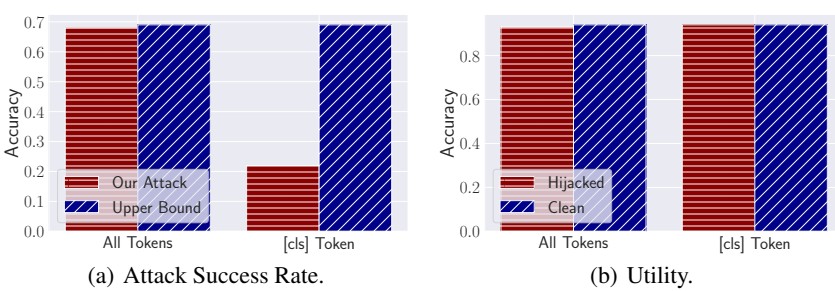

(a) Attack Success Rate.          (b) Utility.

Figure 4: The comparison of our multimodal hijacking attack performances between using complete embeddings of hijacking sentence and the last – "[cls]" – token. The hijacking dataset is Yelp and original dataset is CIFAR-10, while the victim model is ResNet18.

We now evaluate the performance of our attack when using only the "[cls]" token's embedding. To this end, we use the Yelp and CIFAR-10 hijacking and original datasets, respectively. Using the "[cls]" token's embedding only does not change the utility; however, it reduces the attack success rate from 68% to 22%, which clearly demonstrates the advantage of using all embeddings for our attack. We believe using the "[cls]" token's embedding does not change the model's utility (similar to our decision of encoding all tokens as Figure 4 shows), as both approaches result in fused images – with features – that are distinct from the original images. Hence, the target hijacked model is able to learn the original task without any loss of utility.

**Naive Attack:**

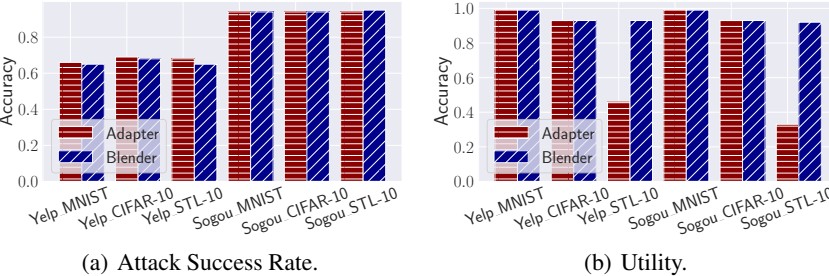

(a) Attack Success Rate.         (b) Utility.

Figure 5: The comparison of our multimodal hijacking attack performances between using the adapter and our Blender. We use x_y notation in the axis to denote the hijacking dataset x and the original dataset y. And the victim model is ResNet18.

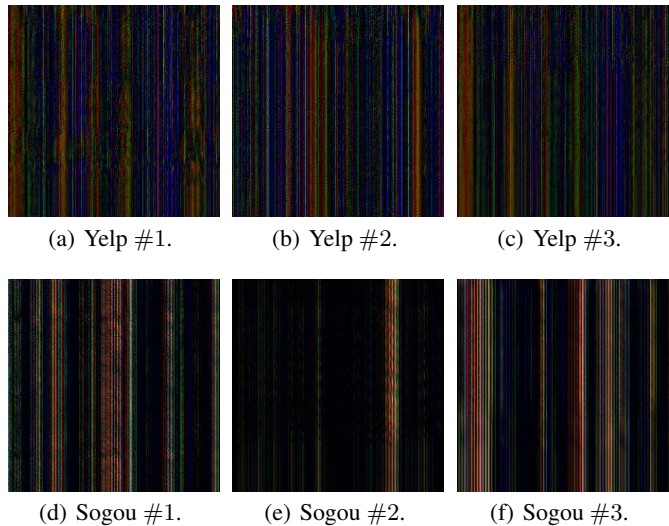

(a) Yelp #1.   (b) Yelp #2.   (c) Yelp #3.

(d) Sogou #1.   (e) Sogou #2.   (f) Sogou #3.

Figure 6: The visual results of the adapter's output using the hijacking sentences of Yelp and Sogou.

We now evaluate the naive approach for the modal hijacking attack, i.e., directly using the output of the adapter instead of the Blender. To this end, we evaluate the naive approach for all of the hijacking and original datasets. The results show that the naive approach can achieve almost the same performance as our attack for some datasets, e.g., CIFAR-10. However, for others the victim models' utility drop significantly. For instance, the utility of the victim models drops to $46\%$ and $33\%$ when using the Yelp and Sogou datasets to hijack an STL-10 classification model. Moreover, we plot the resulting images from the naive approach in Figure 6(a) - Figure 6(f). As the figures show, the resulting images clearly look random and distinct from the original dataset compared to the output of the Blender (Figure 3(a)), which shows the invisibility impact of using the Blender to perform the attack compared to the naive approach.

**Different Feature Extractors:**

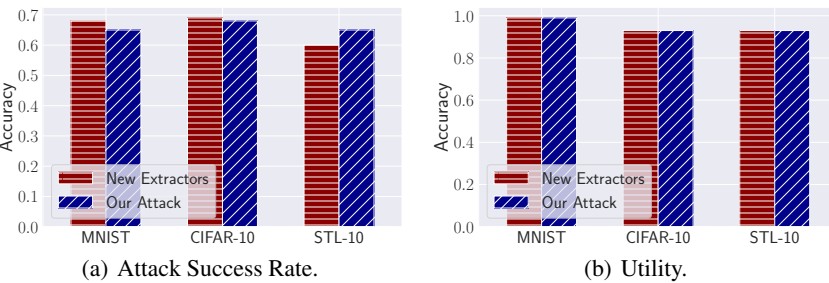

(a) Attack Success Rate.                    (b) Utility.

Figure 7: Our multimodal hijacking attack performance of BART and MobileNetv2 as the – NLP – and – CV – feature extractors. The x-axis denotes the hijacking dataset, while the original dataset is Yelp. And the victim model is ResNet18.

Concretely, instead of using BERT and VGG11 to train the Blender, we use BART (Lewis et al., 2020) and MobileNetv2 (Sandler et al., 2018) as the NLP and CV feature extractors, respectively. The results show that our attack can use different feature extractors depending on the adversary's preference. For instance, it achieves 68%, 69%, and 60% ASR with a negligible drop in utility, when using the Yelp dataset to hijack MNIST, CIFAR-10, and STL-10, respectively.

**Different Victim Models:**

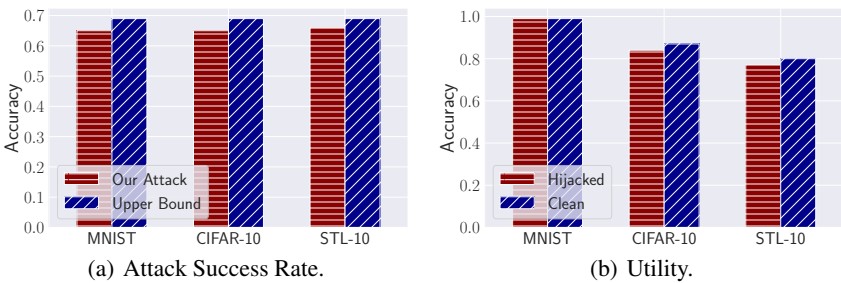

(a) Attack Success Rate.                    (b) Utility.

Figure 8: Our multimodal hijacking attack performance when the victim model is AlexNet. The x-axis denotes the hijacking dataset, while the original dataset is Yelp.

We use the Yelp dataset as the hijacking task for all of the CV tasks while using the AlexNet (Krizhevsky et al., 2012) as the victim model's architecture. As expected, our modal hijacking attack still achieves strong performance against the AlexNet-based models. For example, it achieves 65%, 65%, and 66% ASR with a utility of 99%, 84%, and 77% when hijacking MNIST, CIFAR-10, and STL-10 classification models, respectively. We believe this independence is due to the robust feature extraction of our attack. More concretely, using state-of-the-art language models to extract features from the hijacking datasets results in robust features which when fused with container images, can be picked up by different target models architecture.

**Number of Training Epochs for Blender:**

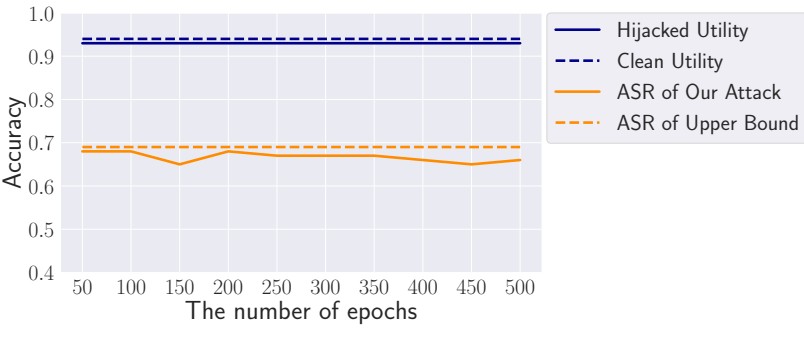

(a) The Attack Performance.

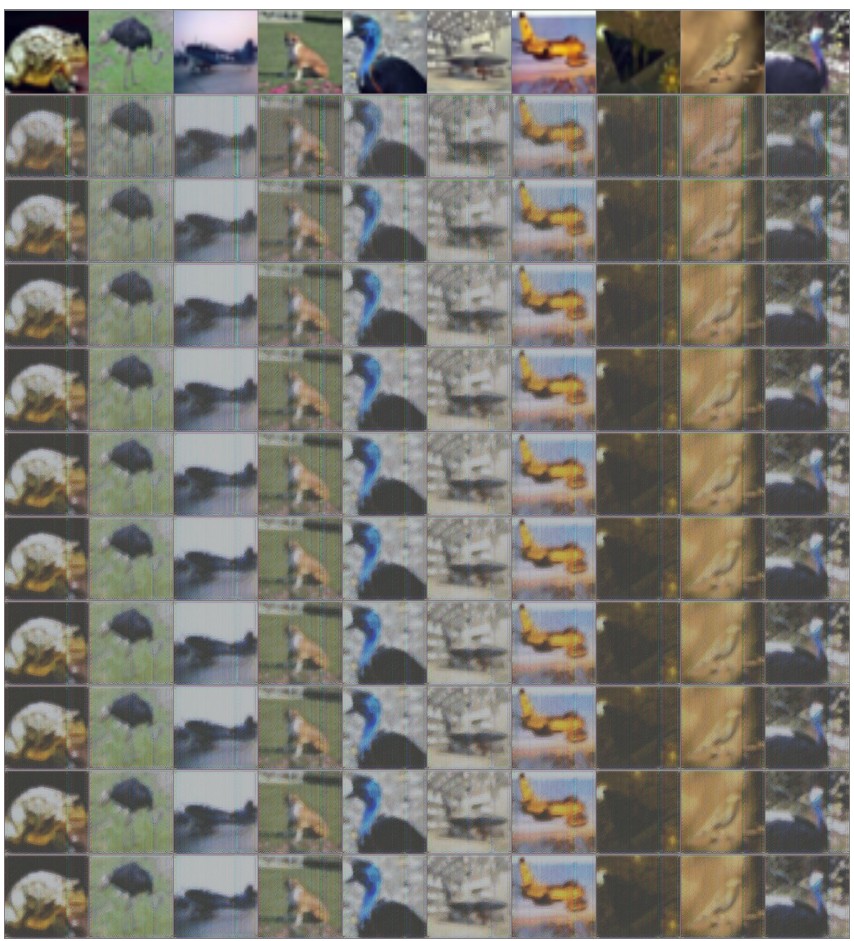

(b) The Visual Results.

Figure 9: Our multimodal hijacking attack performance and the visual results of the Blender's output. The hijacking and original datasets are Yelp and CIFAR-10, while the victim model is ResNet18. In (a), the x-axis denotes the number of epochs which our Blender is trained with. And in (b), the first row presents the visual results of container images, and the following rows indicate the visual results of the output of our Blender trained with different number of epochs, from 50 to 500 with steps of 50.

**Effect of Fine-tuning BERT:**

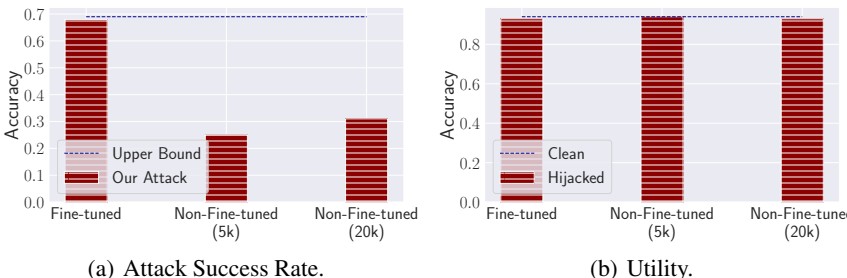

(a) Attack Success Rate.

(b) Utility.

Figure 10: The comparison of our multimodal hijacking attack performances between fine-tuning and not fine-tuning the – NLP – feature extractor of BERT, as well as using a non-fine-tuned BERT with more poisons, i.e., 200,000 fused samples. The hijacking dataset is Yelp and original dataset is CIFAR-10, while the victim model is ResNet18.

We use the Yelp dataset to hijack a CIFAR-10 classification model while using a non-fine-tuned BERT. Our results show that the utility of the victim model does not change; however, the ASR is significantly impacted. More concretely, the ASR drops from $68\%$ to $25\%$. This shows the need to fine-tune $\mathcal{F}_{NLP}$ with the hijacking dataset.

**The Poisoning Rate Effect:**

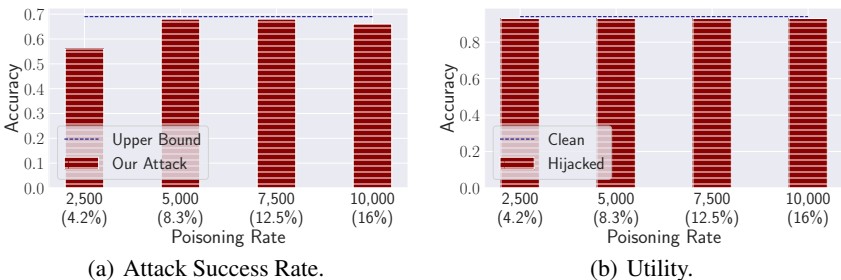

(a) Attack Success Rate.

(b) Utility.

Figure 11: The comparison of our multimodal hijacking attack performances between using different numbers of poisons, from 2,500 to 10,000 with steps of 2,500. The hijacking dataset is Yelp and original dataset is CIFAR-10, while the victim model is ResNet18.

We use the Yelp-CIFAR-10 setting (with 60,000 clean images) and set the number of poisoning – fused – samples from 2,500 (4.2%) to 10,000 (16%) with steps of 2,500 to hijack different models. The results show that using 2,500 samples is too few to hijack the model, i.e., the ASR is only $56\%$. However, increasing the points beyond 5,000 (8.3%) does not improve the ASR. Hence, we use 5,000 hijacked samples for our evaluations in Section 4. On the other hand, the amount of data to poison the dataset here depends on the adversary. As our attack implements a new task, it just depends on how accurate the adversary wants their hijacking task to be. It is also important to note that even if the accuracy is not state-of-the-art, the ability of the model to perform an unethical task (even with low performance) can have the model owner accountable for a violation. And state-of-the-art models usually have millions of inputs, e.g., ImageNet has more than 1.2 M images. Hence, the amount of poisoning data can be negligible depending on the application.

**Reusability of the Blender:**

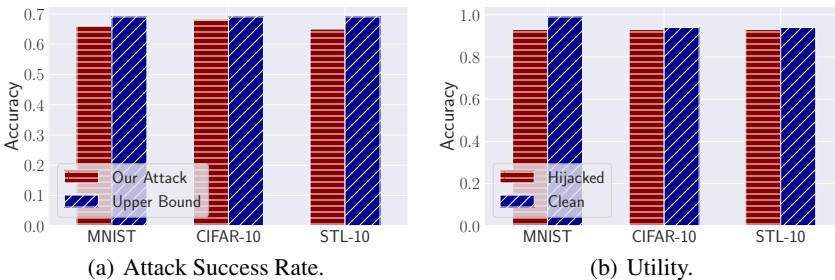

(a) Attack Success Rate.  (b) Utility.

Figure 12: Our multimodal hijacking attack performance when the same Blender is used to fuse the hijacking dataset with different container datasets. The x-axis denotes the container dataset, while the hijacking dataset is Yelp. The Blender is trained on the hijacking dataset of Yelp and container dataset of CIFAR-10. And the victim model is ResNet18.

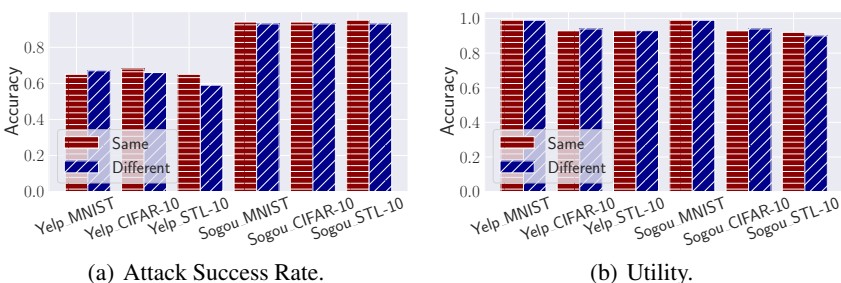

(a) Attack Success Rate.  (b) Utility.

Figure 13: Our multimodal hijacking attack performance when the adversary uses the Blender trained using Yelp(Sogou) but conduct the attack with the hijacking dataset of Sogou (Yelp). The red bars "Same" indicate the case that the Blender is trained on the same dataset as the hijacking dataset. And the blue bars "Different" indicate the case that the Blender is trained on the different dataset from the hijacking dataset. Besides, the x-axis denotes the container dataset, and the victime model is ResNet18.

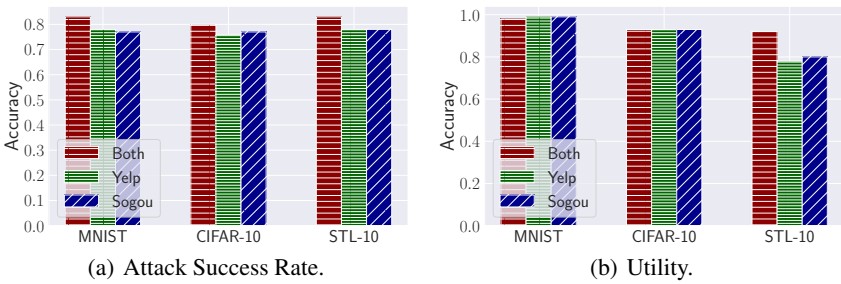

(a) Attack Success Rate.  (b) Utility.

Figure 14: Our multimodal hijacking attack performance when the adversary uses the Blender trained on two dataset (Yelp and Sogou) but conduct the attack with a third hijacking dataset (CoLA). The red bars "Both" indicate the case that the Blender is trained on both Yelp and Sogou but attacks CoLA. The green bars "Yelp" indicate the case that the Blender is trained on Yelp but attacks CoLA. And the blue bars "Blue" indicate the case that the Blender is trained on Sogou but attacks CoLA. Besides, the x-axis denotes the container dataset, and the victim model is ResNet18.

**Container Dataset Creation:**

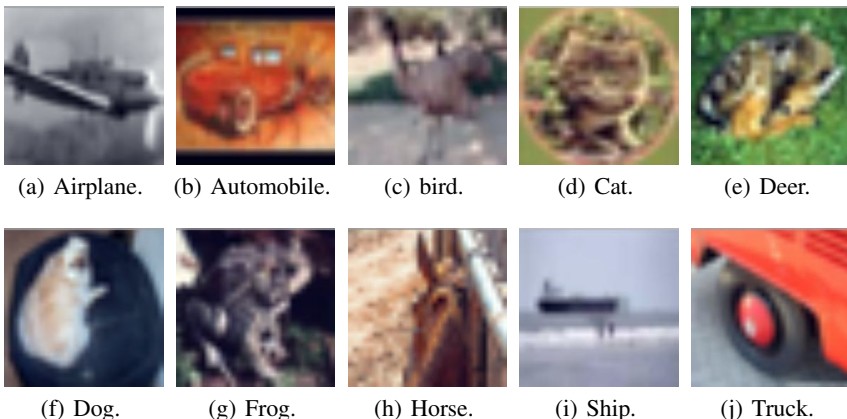

(a) Airplane.    (b) Automobile.    (c) bird.    (d) Cat.    (e) Deer.

(f) Dog.    (g) Frog.    (h) Horse.    (i) Ship.    (j) Truck.

Figure 15: The visual results of examples for each class of CIFAR-10 that are misclassified with the most confidence.

## C  DEFENSE

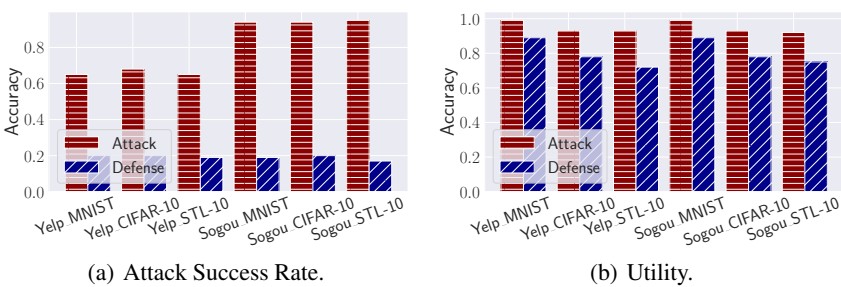

(a) Attack Success Rate.      (b) Utility.

Figure 16: The defense performance against our modal hijacking attack by filtering poisons, where the hijacking tasks are Yelp and Sogou and the original tasks are MNIST, CIFAR-10 and STL-10.

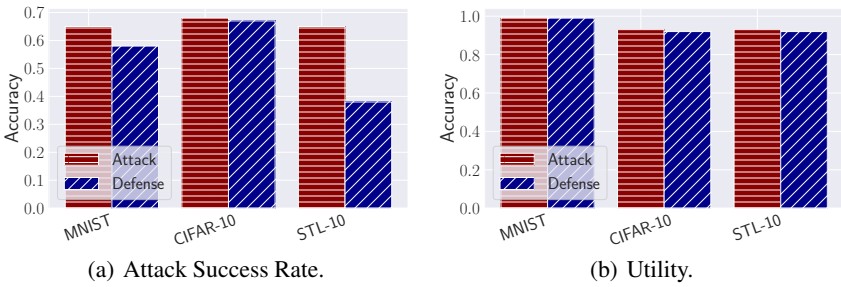

(a) Attack Success Rate.      (b) Utility.

Figure 17: The defense performance against our modal hijacking attack by EPIC proposed by (Yang et al., 2022), where using the three original datasets MNIST, CIFAR-10 and STL-10, Resnet18 as the target model, and Yelp as the hijacking dataset.

# D    ADDITIONAL RESULTS

**More Examples and Explanation:**

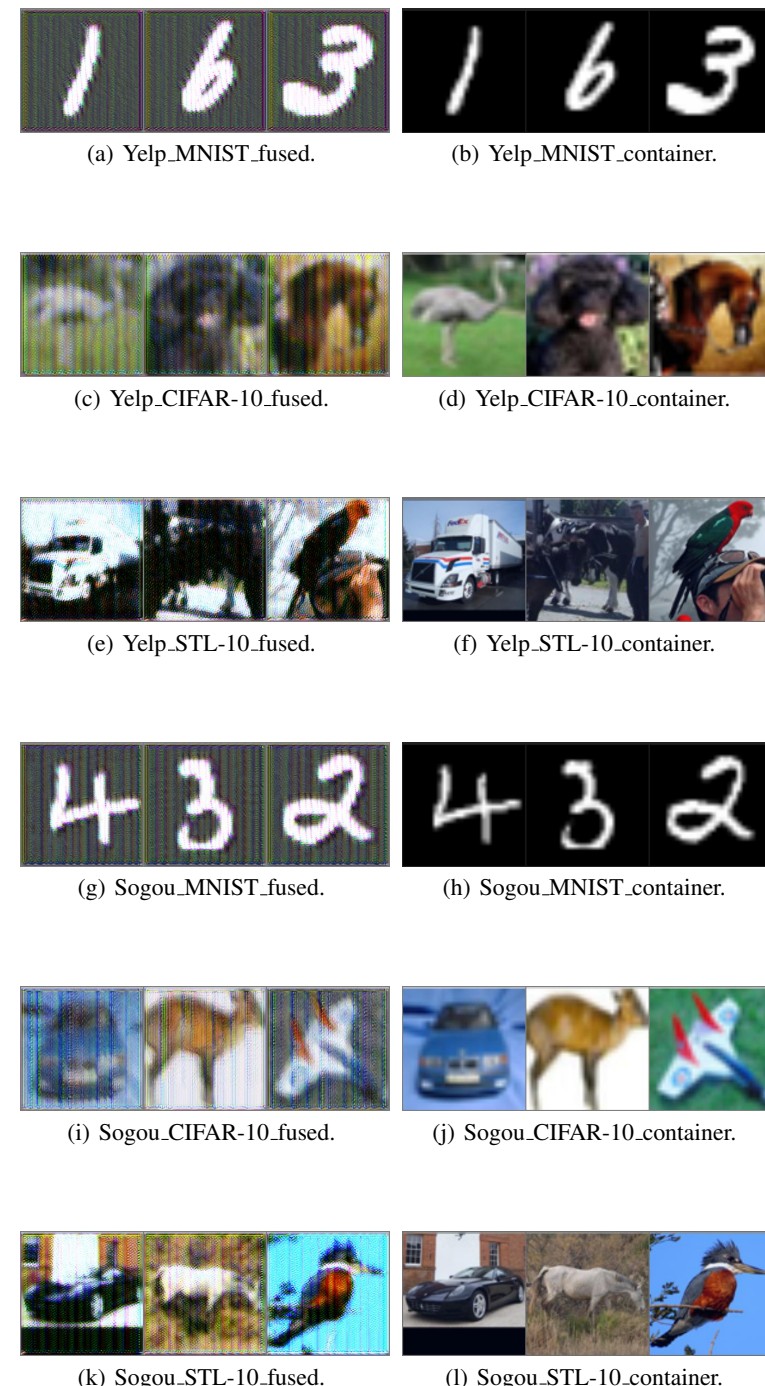

(a) Yelp_MNIST_fused.

(b) Yelp_MNIST_container.

(c) Yelp_CIFAR-10_fused.

(d) Yelp_CIFAR-10_container.

(e) Yelp_STL-10_fused.

(f) Yelp_STL-10_container.

(g) Sogou_MNIST_fused.

(h) Sogou_MNIST_container.

(i) Sogou_CIFAR-10_fused.

(j) Sogou_CIFAR-10_container.

(k) Sogou_STL-10_fused.

(l) Sogou_STL-10_container.

Figure 18: The visual results of our Blender's output and corresponding container images. We use x_y_z notation to denote the hijacking dataset x and the container dataset y, while z indicates the image type, i.e., a fused or container image.

Table 1: The details of Figure 18, where we display the fused and corresponding container images. The hijacking datasets include Yelp and Sogou while the container datasets contain MNIST, Cifar-10 and STL-10. Regarding the fused images, we assign labels according to the NLP sentences. For instance, when the ground truth of a given NLP sentence is "5 stars", the label of "Digit 4" will be assigned to all fused MNIST-like images that contain the NLP feature of this sentence. Besides, we also depict the corresponding container images with their labels.

| Hijacking dataset | Container dataset | Fused | | | | Container | |
|---|---|---|---|---|---|---|---|
| | | Image | Image label | NLP sentence | NLP ground truth | Image | Image label |
| Yelp | MNIST | | Digit 4 | "Some of the best chow around–love this place. The bread and salads and soups are great." | 5 stars | | Digit 1, 6, 3 |
| | Cifar-10 | | Airplane | "The worst dental office I ever been. No one can beat it!!! You should avoid it at any time." | 1 star | | Bird, Dog, Horse |
| | STL-10 | | Bird | "Far away from real Chinese food. Doesn't even taste good as American style Chinese food." | 2 stars | | Truck, Horse, Bird |
| Sogou | MNIST | | Digit 3 | "zho1ng hua2 ju4n jie2 FRV ya4o shi" | Automobile | | Digit 4, 3, 2 |
| | Cifar-10 | | Airplane | "2008nia2n 5 yue4 19 ri4 , be3i ji1ng go1ng ye4 da4 xue2 ti3 yu4 gua3n shi4 be3i ji1ng..." | Sports | | Automobile, Deer, Airplane |
| | STL-10 | | Bird | "ya1n zha4o du1 shi4 ba4o ju4 hu4 she1n zhe4ng qua4n jia1o yi4 suo3 go1ng ka1i xi4n..." | Finance | | Car, Horse, Bird |

**Evaluation on Tiny ImageNet:**

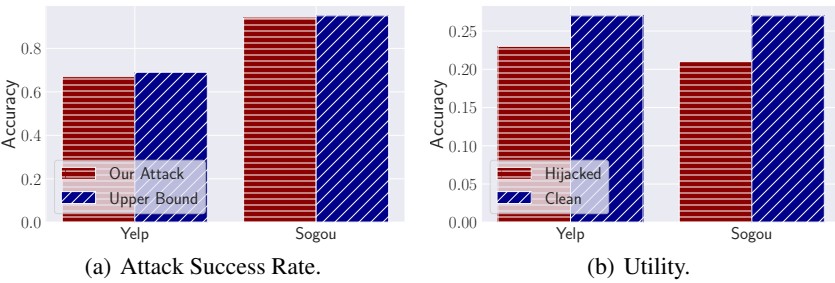

(a) Attack Success Rate.  (b) Utility.

Figure 19: Our multimodal hijacking attack performance. We use the x-axis to denote the hijacking dataset. And we use the Tiny ImageNet as our original dataset, mobilenetv2 as the target model. The low utility of the clean models (27%) is due to a large number of labels (1,000), and the limited number of samples in the dataset.

Table 2: The visual results and explanation of the Blender's outputs using Yelp and Sogou as the hijacking datasets while Tiny Imagenet as the container dataset. Regarding the fused images, we show the images and their labels. Besides, we also list the embedded NLP hijacking sentences and corresponding ground truth, e.g., "Wast there last Friday. Seats right in front if the stage. The show was good. The headliner..." with the label of "4 stars". Regarding the container images, we show the images with their labels.

| Hijacking dataset | Container dataset | Fused | | | | Container | |
|---|---|---|---|---|---|---|---|
| | | Image | Image label | NLP sentence | NLP ground truth | Image | Image label |
| Yelp | Tiny | | Rocker | "Wast there last Friday. Seats right in front if the stage. The show was good. The headliner..." | 4 stars | | Bathtub, Teddy, School bus |
| Sogou | ImageNet | | Lemon | "te2ng xu4n ke1 ji4 xu4n be3i ji1ng shi2 jia1n 5 yue4 16 ri4 xia1o xi1 , ju4 guo2 wa4i..." | Technology | | Miniskirt, King penguin, Broom |

# E BLENDER DETAILS

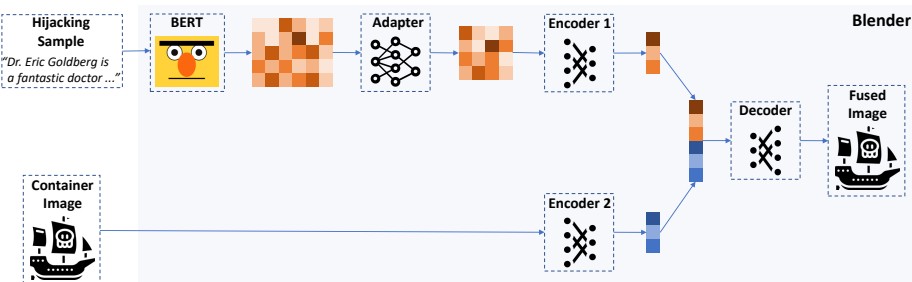

Figure 20: The pipeline of our blender.

