# OpenReview forum: "Vera Verto: Multimodal Hijacking Attack"
_ICLR.cc/2023/Conference — Submitted to ICLR 2023_

### Official Review · Reviewer_GKEm · 2022-10-21

**Confidence:** 3
**Correctness:** 3
**Technical Novelty And Significance:** 3
**Empirical Novelty And Significance:** 3
**Recommendation:** 6

**Clarity, Quality, Novelty And Reproducibility:**

The paper is clear, and the ideas are novel
Reproducibility can not be assessed since no code was submitted.

**Strength And Weaknesses:**

**Strengths.**
+ proposing a shift of reprogramming towards different domains is very interesting, since it opens different possibilities (i.e. reroute the computational power of image classifier for other tasks, like threat hunting for instance)
+ the methodology works by just considering a dataset similar to the one used for training, and its application is realistic
+ experiments are rich, and they convey the effectiveness of the methodology

**Weaknesses.**
+ **Not really multi-model.** While the paper focuses on proposing a contribution on the multi-modality of hijacking attacks, the authors only focus on translating an NLP problem to a CV one. While this is not bad per-se, the author should better clarify how such methodology could potentially be general to any domain. One example would be specifying which are the components that need to be implemented in order to port the hijacking task to another domain (audio, malware detection, etc). Moreover, I imagine that there can be domains where hijacking is much more difficult, or much easier.
+ **Presentation can be improved.** While the paper reads well, the overall presentation could be improved: most of the more interesting figures are put aside in the appendix, while only Fig.2 and 3 are put into the main paper. Also, the paper could contain more examples of NLP sentences / label and corresponding fused images to better understand the effects of the techniques (Fig.3 only contains the images, but not the predicted class of the fused image and the ground truth of the NLP sample)

**Summary Of The Paper:**

The authors propose Blender, an adversarial reprogramming technique (or hijiacking) that converts NLP samples inside legitimate images of a container dataset to use computational power of the victim model to solve the task decided by the attacker.
Blender is trained to minimise the summation of two objective functions, namely one that takes into account the fused image and the container one (visual loss), and one that force the learned feature representation of the network of NLP tokens and images to match (semantic loss). The authors show that the hijacking is successful in different combinations of container dataset of text and images, by highlighting that the predictive accuracy is matching the accuracy on the original data.

**Summary Of The Review:**

Hijacking can be applied cross-domain, in particular between NLP and CV tasks by creating a model that learns how to apply the translation, which is the core contribution.
However, the authors should better discuss such cross-domain generalisation, and help the reader understand how this can work on other different couples of domains.

---

> ### Author Response · Authors · 2022-11-09
> **Response to Reviewer GKEm**
>
> # Regarding the multi-modality:
> We thank the reviewer for pointing this out. Indeed, adding a discussion about this would improve the paper. Intuitively, our technique can be adapted for different scenarios where the hijacking task has feature extractors, e.g., language models for the NLP tasks. However, different modalities have different difficulties, i.e., how easy/hard the original data can be modified. For instance, consider the opposite direction of implementing a CV task into an NLP model. It would be significantly difficult/different, as text inputs are not continuous, and hence cannot be changed as the images do.
>
> # Regarding presentation
> We are limited by space restrictions; however, we will add more sentence-image pairs (in the appendix) and add to Fig.3 the original and predicted labels.
>
>
> **We will add a section to discuss our methodology and how it can be extended to other modalities and add more examples in the revised version of the paper.**

---

### Official Review · Reviewer_dqpP · 2022-10-24

**Confidence:** 3
**Correctness:** 2
**Technical Novelty And Significance:** 2
**Empirical Novelty And Significance:** 2
**Recommendation:** 5

**Clarity, Quality, Novelty And Reproducibility:**

#### **Clarity**

- **C1. Evaluating metric.** In Sec. 4.2.2. the description of *attack success rate* is hard to follow. How do you define *a clean testing dataset (for the hijacking task)*? What do you mean by *train an NLP classification model for each hijacking task*? What are the input, output, and target (label) of the NLP classification model? How do you compare the performance with the victim model on the hijacking task, and with whom? This subsection deserves to be more specific and accurate for readability and assessing the merit of the work. *It would be helpful to provide the revised version of this subsection to reassess this issue.*

- **C2. Vague definition.** In the first two paragraphs of the introduction, the definition of the model hijacking attack is unclear. Figure 1, with a shallow description, is frustratingly hard to understand.


#### **Quality**

- In-text citation is wrongly used for the parenthetical citation.

#### **Novelty**

- Compared with Salem et al. (2022a), this work seems to be incremental to multimodal hijacking attacks while the modality of the hijacking sample is changed along with the corresponding encoder. The authors should provide notable speculations or discussions in the extension to multimodal setting, in addition to the argument that it generalizes unimodal tasks. If the adversary wants to attack, why do they want to attack with the other modality if it underperforms the same modality attacks?

**Strength And Weaknesses:**

#### **Strength**

- It generalizes the previous unimodal setup, the model hijacking attack (Salem et al., 2022a), to the modal hijacking attack.
- The extensive studies on hyperparameters and design choices in Sec. 4.4.
- It is appreciated to include the discussion on a defense strategy and its experimental results.

#### **Weakness**

- **W1. Simple baselines.** For the model comparison, this work does not provide the performance of baselines excluding the adapter-only blender (Sec. 4.4). One simple baseline would be the container samples having random labels without decoding. This baseline would show the blender's effectiveness, excluding the effect of randomly-assigned labels. The other baseline would be training with original and container samples with ground-truth labels (or nearest neighbor labels). This baseline gives the hijacking ability only with the data distributional changes since the adversary would attack with the container distribution (Sec. 3.4).

- **W2. Bi-model and unidirectional.** This work only provides the experiment on bi-modal unidirectional settings. Can this model hijack an NLP-based target model with a CV-based task?

- **W3. The motivation of NLP-based tasks.** The hijacking dataset is randomly mapped to a container image. Why don't we semantically map exploiting NLP-based tasks? For example, the hijacking sample is mapped to the nearest neighbor label in the common embedding space (while the labels are embedded to the same space)?

**Summary Of The Paper:**

They proposed a transformation to multimodal extension from the model hijacking attack by Salem et al. (2022a). This work takes a data poisoning approach while the fused dataset is used to poison a victim model. The evaluation metrics are the attack success rate and utility, where both metrics are important to attain simultaneously to hijack the victim model successfully.


**Summary Of The Review:**

The motivation of this work (W2, W3) and clarity (C1, C2) issues hinder recommending this paper to the conference. The comparison with more diverse baselines (W1) would strengthen the validation of the idea.

---
After reading the author's feedback, the shared concerns of the lack of justification and effective validations remain. I will hold the current evaluation.

---

> ### Author Response · Authors · 2022-11-09
> **Response to Reviewer dqpP**
>
> # Regarding Baselines (W1):
> We present the results against the baseline in Section 4.4 as it is a naive attack that does not need the Blender. We thank the reviewer for proposing different baselines. We are very open to evaluating our attacks against different baselines; however, we would appreciate it if the reviewer can provide more details, as we are a little confused regarding the two proposed baselines. Just for clarification, we do not perform any random label assigning except for determining a one-to-one mapping for the hijacking tasks and the original one (of course, an adversary can propose a different mapping, we are just proposing a straightforward one). This means the same container image can be fused with hijacking samples of different labels; hence, the model's output will be different for the same container image depending on which hijacking sample it was fused with. However, the container image alone does not have to do anything with the hijacking task.
>
> # Regarding Bi-model and unidirectional (W2):
> Indeed this would be an interesting next step to hijack an NLP model with a CV task. However, it would need a different technique, hence we leave it for future work.
>
> #  Motivation of NLP-based tasks (W3):
> Indeed this would be an interesting way of selecting container images. Our attack is independent of how container images are selected. However, there are a couple of drawbacks to this approach. The first is that label mapping is done with respect to the whole classes in the hijacking task, i.e., each hijacking class needs to be consistently mapped to a distinct label from the original task, to be able to revert this mapping after the hijacked model is deployed (in the testing/evaluation phase). In other words, we cannot label every fused image with the same hijacking label to a different label. It is also important to note that the embedding space of the hijacking task and the original one is distinctly different due to the difference in data types. Moreover, if we select the nearest container image while training the model, then we would also need to repeat this prosses after the model is deployed, which would require all users of the hijacked model to have access to all candidate images.
>
> Another approach is to consider candidate images that are confusing to humans, i.e., images that are on the boundary between classes. By using these images, the attack gets more stealthy since even if a human looks at them, they would not be confident to determine their true label.
>
> # Evaluating metric (C1)
> ## Regarding the clean testing dataset:
> Intuitively, the clean testing dataset is the testing dataset of the original CV model, i.e., from the same distribution as the original dataset but not used when training. We use this dataset to make sure the utility of the hijacked model is not jeopardized.
> ## Regarding training an NLP classification model for each hijacking task:
> To compare how successful our model hijacking attack is. We need to evaluate how it performs on the hijacking task. To this end, we train an NLP model on the hijacking dataset. This model is trained with the hijacking dataset as is, i.e., the data and labels of the hijacking dataset are not changed. These NLP classification models represent if the adversary wants to perform the hijacking task without the modal hijacking attack; hence, it represents the upper bound of the attack, i.e., the closer the performance of our modal hijacking attack to these models, the better the attack is. We compare the performance of these NLP models and the hijacking models using a testing dataset from the same distribution as the hijacking dataset. Of course, for our attack, we need to first fuse the testing dataset with container images by our trained Blender.
>
> # Vague definition (C2):
> We will clarify the definition of the model hijacking attack for better understanding.
>
> # Citation:
> We thank the reviewer for pointing this out and we will fix the citations.
>
> # Novelty:
> Please check our first response to Reviewer 2T2w. Moreover, the adversary does not control the original task to select a unimodal or multimodal setting. Having a multimodal attack increases the scope for the adversary, so they do not necessarily need to find a model with an original task from the same modality as their hijacking task. In other words, this new attack makes it easier for the adversary to find a model to hijack.
>
> **We thank the reviewer again for the multiple suggestions and will incorporate them in the next version of the paper for better readability.**

---

### Official Review · Reviewer_2T2w · 2022-10-24

**Confidence:** 4
**Clarity, Quality, Novelty And Reproducibility:** The writing clarify and quality are g…
**Correctness:** 4
**Technical Novelty And Significance:** 3
**Empirical Novelty And Significance:** 3
**Recommendation:** 5

**Strength And Weaknesses:**

Strength:
- The paper investigates an exciting setting for model hijacking attacks.

Weaknesses:
- At a technical level, the proposed technique is similar to the original model hijack attack on images proposed by Salem et al. Both employ an encoder-decoder structure and similar ideas of making the fused sample close to the hijacking sample in the feature space. The technical novelty needs to be further justified.
- The detectability of the proposed attack is not sufficiently evaluated. In fact, there have been many recent advances in detecting poisoned instance; to name a few, [1], [2], [3]. It is possible that the poisoned instances can be easily detected in the frequency domain.

[1]: Friendly Noise against Adversarial Noise: A Powerful Defense against Data Poisoning Attacks. NeurIPS 2022.
[2]: Not All Poisons are Created Equal: Robust Training against Data Poisoning. ICML 2022.
[3]: Rethinking the Backdoor Attacks' Triggers: A Frequency Perspective. ICCV 2021.


**Summary Of The Paper:**

The paper proposes a technique to enable model hijacking attacks for data with different modalities. Specifically, it makes the hijacked model predict a image-related label to a given text.

**Summary Of The Review:**

Overall, the paper studies an interesting attack setting for model hijacking attacks ---how to use the victim model from a domain to serve tasks in another domain. However, the paper can be further improved by clarifying the technical novelty, i.e., the difference from Salem et al and why these differences are significant. Moreover, the detectability of such an attack needs to be further discussed.

---

> ### Author Response · Authors · 2022-11-09
> **Response to Reviewer 2T2w**
>
> # Regarding the difference to the model hijacked attack proposed by Salem et al.:
> The model hijacking attack focuses on a single type of data, unlike our modal hijacking one which expands the scope to a multi-modal setting. The challenge that stems from that change is the needed transformation from a discrete domain (Natural Language Processing) to a continuous one (Computer Vision). We believe it is not trivial to make this transformation; for instance, we show in Section 4.4 that using just the outputs of a langue model would not result in a good performance (more generally, we demonstrate what is needed for this transformation to work). To the best of our knowledge, the modal hijacking attack is the first work to combine different data modalities, which increases the scope of the possible applications for the hijacking attack, e.g., the adversary can hijack unethical NLP tasks into CV models. Moreover, we believe this work can encourage the exploration of the applicability of performing the attack for different modalities (which can also be with the benign aim of model compression). Finally, our modal hijacking attack is more general, i.e., the Blender can be applied with different hijacking and original datasets. Hence, it is cheaper for the adversary to hijack target models as shown in the “Reusability of the Blender” paragraphs in Section 4.4.
>
> # Regarding defenses:
> We try different defenses in Section 4.5, which shows that our Modal hijacking attack cannot be defended against without a significant drop in the utility. We thank the reviewer for pointing out different defenses. [1] is just accepted in NeurIPS this year and does not have a public code to compare its performance with our attack. [2] assumes targetted poisoning attacks with limited perturbations (which we do not); However, **we are currently trying to reimplement/adapt their code and try it on our attack.** Finally, [3] proposes an interesting defense using the frequency domain; however, it also proposes a smoother attack that our modal hijacking attack can easily adapt to, i.e., to have a smoother frequency domain even if the original attack does not.
>
> **We will clarify all of these points in the next version of the paper (which will be uploaded after the experiments are finished).**

---

> > ### Author Response · Authors · 2022-11-10
> > **Results for the additional defense (from [2])**
> >
> > We adapted the code for [2] and evaluated our hijacking attack against it using the three original datasets MNIST, CIFAR-10 and STL-10, Resnet18 as the target model, and Yelp as the hijacking dataset. Our results show that the defense is able to reduce the attack success rate, however, our hijacking attack is still effective. We summarize the results in the table below:
> >
> > | Dataset| Utility Drop|ASR Drop| ASR After Defense| ASR  Before Defense|
> > | ----------- | ----------- |----------- |----------- |----------- |
> > | MNIST| 0%| 7%|58%|65%|
> > | CIFAR-10| 1%|1%|67%|68%|
> > | STL-10| 1%|27%|38%|65%|

---

### Official Review · Reviewer_TtX4 · 2022-10-25

**Confidence:** 3
**Correctness:** 4
**Technical Novelty And Significance:** 3
**Empirical Novelty And Significance:** 4
**Recommendation:** 5

**Clarity, Quality, Novelty And Reproducibility:**

The paper is easy to read and follow. It is novel but I have some major concerns on its motivation, practicality, and the root cause analysis.

**Strength And Weaknesses:**

Strength:
- Propose the first model hijacking attack in which the adversary can hijack a CV-based targeted model by an NLP-based task
- The evaluation shows the high effectiveness of their attack and include multiple design choice in evaluation

Weakness:
- Lack of the justification about the multimodal settings.
There is no justification about their multimodal settings. For instance, why do you select NLP plus CV as a case study? Does these combination is representative for the new parties to the training pipeline mentioned in the paper? In the paper, the authors only mention that ML has achieved great success in many domains and hijacking task might deal with data from other modalities. But there is no justification or support. Without such justification, it is unclear whether the problem studied in this paper is a real problem or not. At least, the authors should provide more details about the representativeness of their multimodal settings.

- Lack of practical evaluation.
This paper is motivated by a real-world new parties to the training pipeline, such as users who contribute training data and companies that provide computing resources. However, later on there is no evaluation on a practical systems but only evaluate on small dataset such as MNIST, CIFAR-10, and STL 10, in which all the images are of low resolution. It is unclear whether their novel methodology can work in a large image dataset and how much cost it will be. Also, the high-resolution artifacts in the fused image may be also very easy to be detected. Without such evaluation, it is very difficult to judge whether their attack can indeed work in real-world system or at least in real-world high resolution image setting. Also, considering the poison rate effects, although the authors evaluate on different settings, there is no justification/supports on which number is practical/reasonable for a real-world system. If their poisoning rate is higher than that the practical threshold, their design may be empirically flawed.

- Lack of detailed reasoning on some design choices.
In evaluation section 4.4, there generally lack of detailed reasoning on some design choices. For instance, why the model hijacking attack is independent of the victim's model architecture; why "[cls]" token's embedding only does not change the utility. In this section, the authors only list out the experimental results but do not provide detailed analysis especially on the root cause part. If more details can be discussed, it can benefit the future research designs and make this paper more insightful and interesting.

**Summary Of The Paper:**

In this paper, the authors transform the model hijacking attack into a more general multimodal settings, where the hijacking and original tasks are performed on data of different modalities. Specifically, they focus on the setting where an adversary implements a natural language processing (NLP) hijacking task into an image classification model. Their evaluation results show the high effectiveness of their attack.

**Summary Of The Review:**

I believe this paper is slightly below the the acceptance threshold.

I have some major concerns as follows:

- Lack of the justification about the multimodal settings.

- Lack of practical evaluation.

- Lack of detailed reasoning on some design choices.

---

> ### Author Response · Authors · 2022-11-09
> **Response to Reviewer TtX4**
>
> # Regarding selecting NLP and CV:
> We focus on NLP and CV as these are the most used ML applications, e.g., the multiple available translators such as DeepL and Google Translate, and the different face detectors on social media platforms. The new parties included in the training dataset are due to the amount of needed data to train the new state-of-the-art models, e.g., the recent GitHub Copilot (https://github.com/features/copilot) is trained on publicly available open source codes (which to poison such a model an adversary would just need to publish their own poisoned code and wait for it to be used in training). Combining different modalities removes the restriction for the adversary to find a target model with strictly the same data type as their hijacking task; hence, we believe it significantly simplifies the assumptions of the model hijacking attack. We will also add a discussion about extending our attack to different setups, please check our first response to Reviewer GKEm for more details.
>
> # Regarding lack of practical evaluation
> ## Regarding the used datasets:
> We are just showcasing our technique using these three datasets; however, we believe our attack is generalizable to higher-quality images. **We are currently running an experiment on a subset of ImageNet (to reduce the computation time we are not using the full ImageNet) to demonstrate the effectiveness of our attack.**
>
> ## Regarding the artifacts:
> Indeed there exist some artifacts on the images; however, as we show in Section 4.5, they do not make our technique easier to detect. We aim with this work to demonstrate the possibility of hijacking models with a totally different data type and believe reducing the artifacts is an interesting direction for future works, e.g., by introducing a distinguisher to the pipeline. Finally, we only stop the training of the blender at 50 epochs however, increasing it to 200 reduces the artifacts as shown in Figure 9.
>
> ## Regarding the poisoning rate:
> The amount of data to poison the dataset here depends on the adversary. As our attack implements a new task, it just depends on how accurate the adversary wants their hijacking task to be. It is also important to note that even if the accuracy is not state-of-the-art, the ability of the model to perform an unethical task (even with low performance) can have the model owner accountable for a violation. Finally, state-of-the-art models usually have millions of inputs, e.g., ImageNet has more than 1.2 M images. Hence, the amount of poisoning data can be negligible depending on the application.
>
> # Regarding design choices
> ## Regarding the independence of the target model:
> We believe this independence is due to the robust feature extraction of our attack. More concretely, using state-of-the-art langue models to extract features from the hijacking datasets results in robust features which when fused with container images, can be picked up by different target models architecture.
>
> ## Regarding the "[cls]" token's embedding:
> We believe using the  "[cls]" token's embedding does not change the model's utility (similar to our decision of encoding all tokens as Figure 4 shows), as both approaches result in fused images – with features – that are distinct from the original images. Hence, the target hijacked model is able to learn the original task without any loss of utility.
>
> **We will clarify all of these points in the next version of the paper (which will be uploaded after the experiments are finished).**

---

> > ### Author Response · Authors · 2022-11-10
> > **Results for high quality images (ImageNet)**
> >
> > We use the Tiny ImageNet as our original dataset, mobilenetv2  as the target model, and Yelp and Sogou as the hijacking dataset. Our experiments show that the utility of the model drops by only 4%  and 6% (from 27% to 23% (21%)), while the ASR is almost the same (the drop is less than 2%) for the Yelp and Sogou hijacking tasks, respectively. This result shows that our technique is generalizable to higher-quality images. The low utility of the clean models (27%) is due to the large number of labels (1,000) and the limited number of samples in the dataset.

---

### Author Response · Authors · 2022-11-09
**Common response to all reviewers**

We would like to thank the reviewers for their detailed insightful feedback. We will make all requested editorial changes and address the raised questions with respect to each review.

---

### Author Response · Authors · 2022-11-18
**The uploading of the revised version of paper**

We thank the reviewers for the suggestions to help improve our paper and have uploaded a revised version of the paper addressing all the comments (in blue).

In summary:
- In Section 1, we added more clarification of the motivation, novelty, and justification of the multimodal settings.
- Regarding the practical evaluation, we evaluated our attack on Tiny ImageNet as Section 4.3 shows, and full results and examples with details are depicted in Figure19 and Table 2 (Appendix D).
- Regarding the design choices, we added more explanation in Section 4.4 and Appendix B.
- Regarding the defense, we evaluated another method, i.e., EPIC, in Section 4.5 and show the full results (that it cannot fully defend against our attack) in Figure 17 (Appendix C).
- Regarding the evaluating metric, we added more description about the involved terminologies in Section 4.2.2.
- Regarding the presentation, we added more sentence-image pairs, and the original and predicted labels in Figure 3, Table 1, and Table 2 (Appendix D).
- Regarding the vague definition, we added more description to Figure 1 and added a pipeline of our Blender in Figure 20 (Appendix E).
- Finally, we also discussed the applicable multimodal scenarios and limitations of our work in Appendix A.

We thank again the reviewers and hope this revised version could provide a better understanding of our work.

---

### Decision · Program_Chairs · 2023-01-20

**Decision:**

Reject

**Justification For Why Not Higher Score:**

The method has limited novelty, and the empirical results are not strong enough.

**Justification For Why Not Lower Score:**

N/A

**Metareview: Summary, Strengths And Weaknesses:**

The paper proposes the model hijacking attack from the computer vision domain to the multi-modal setting (NLP-vision). However, a majority of the reviewers think the novelty is limited since the proposed method is a simple extension of (Salem et al., 2022a). Given the limited novelty, the reviewers think that the current empirical results are not strong and comprehensive enough for this paper to publish in ICLR.